# Lactoferrin—The Health-Promoting Properties and Contemporary Application with Genetic Aspects

**DOI:** 10.3390/foods12010070

**Published:** 2022-12-23

**Authors:** Anna Jańczuk, Aneta Brodziak, Tomasz Czernecki, Jolanta Król

**Affiliations:** 1Department of Quality Assessment and Processing of Animal Products, Faculty of Animal Sciences and Bioeconomy, University of Life Sciences in Lublin, Akademicka 13, 20-950 Lublin, Poland; 2Department of Biotechnology, Microbiology and Human Nutrition, Dietitian Service, Faculty of Food Science and Biotechnology, University of Life Sciences in Lublin, Skromna 8, 20-704 Lublin, Poland

**Keywords:** lactoferrin, bioactive potential, nutrigenomics, nutrigenetics, obesity, type 2 diabetes, cardiovascular diseases, personalized nutrition, food industry

## Abstract

The aim of the study is to present a review of literature data on lactoferrin’s characteristics, applications, and multiple health-promoting properties, with special regard to nutrigenomics and nutrigenetics. The article presents a new approach to food ingredients. Nowadays, lactoferrin is used as an ingredient in food but mainly in pharmaceuticals and cosmetics. In the European Union, bovine lactoferrin has been legally approved for use as a food ingredient since 2012. However, as our research shows, it is not widely used in food production. The major producers of lactoferrin and the few available food products containing it are listed in the article. Due to anti-inflammatory, antibacterial, antiviral, immunomodulatory, antioxidant, and anti-tumour activity, the possibility of lactoferrin use in disease prevention (as a supportive treatment in obesity, diabetes, as well as cardiovascular diseases, including iron deficiency and anaemia) is reported. The possibility of targeted use of lactoferrin is also presented. The use of nutrition genomics, based on the identification of single nucleotide polymorphisms in genes, for example, FTO, PLIN1, TRAP2B, BDNF, SOD2, SLC23A1, LPL, and MTHFR, allows for the effective stratification of people and the selection of the most optimal bioactive nutrients, including lactoferrin, whose bioactive potential cannot be considered without taking into account the group to which they will be given.

## 1. Introduction

The growing interest among consumers in recent years in food products that have a positive effect on the human body proves conscious concern for health and well-being. However, this imposes the introduction of changes in the food industry related to the implementation of innovative technologies in food production. One such industry is the dairy industry. Milk proteins, including lactoferrin, have gained particular importance in recent years. Nowadays, it is beginning to move towards profiling products and adjusting their properties and compositions to the requirements of specific consumer groups, also taking into account profiling for genetic predisposition.

The aim of the study is to present a review of literature data on lactoferrin’s characteristics, applications, and multiple health-promoting properties, with special regard to nutrigenomics and nutrigenetics. This has not yet been the subject of any review devoted to lactoferrin. Generally, nowadays, the combination of food production with diet therapy or nutritional prophylaxis is a niche, especially with the use of advanced nutrigenomic and nutrigenetic tools. In the future, it will be a standard procedure due to the increasing demands of consumers, as well as the decreasing prices of genetic tests, the increased availability of these tests, and also an increase in the knowledge of specialists in the field of human nutrition and dietetics.

## 2. Materials and Methods

A comprehensive search for scientific articles on the reviewed topic devoted to lactoferrin was carried out using databases such as Scopus, Science Direct, Springer, PubMed, Medline, and Web of Science. When searching for information, the following keywords were used: “lactoferrin”, “activity”, “production”, “pharmaceutical products”, “cosmetics”, “obesity”, “type 2 diabetes”, “cardiovascular diseases”, “anti-inflammatory effect”, “immune system”, “nutrigenomics”, “nutrigenetics”, and “personalized nutrition”. This study includes only those articles in which the data from the obtained research were related to bovine lactoferrin. The search was mainly limited to reports in the English language. Based on the criteria and keywords listed, 106 reports (including 85 studies) were selected for their findings. The review included works published in the years 2002–2023.

## 3. Occurrence and Structure of Lactoferrin

Lactoferrin (LF) is a protein belonging to the transferrin family, discovered for the first time in cow milk in 1939. It is produced by the epithelial mucosa. It is present in milk and colostrum, but can also be found in tears, saliva, gastric mucosa, the spleen, lymph nodes, skin, and even white blood cells [1,2,3]. LF accounts for about 1% of whey proteins [1,3]. Its concentration is highest in colostrum, but the level is not constant and decreases over the course of lactation, reaching its lowest values in mature milk [4]. The LF concentration is lower in cow milk than in human milk. In bovine colostrum, it is 2–5 mg/mL but decreases sharply during lactation to a level of 0.1–0.3 mg/mL in mature milk. The LF concentration in milk depends on genetic factors such as species or breed, as well as non-genetic factors, including parity and stage of lactation, daily milk production, and udder health [5,6,7]. The LF concentration in human serum is usually lower (about 1000 ng/mL) but increases with the degranulation of neutrophils during infection [8,9].

LF consists of about 700 amino acid residues, with a molecular weight of about 78 kDa [1]. Bovine lactoferrin (bLF) consists of 696 amino acids, while human lactoferrin has 691. Its isoelectric point is fairly high (pI ≈ 8.7). The saturation of LF with iron (as one of the transferrins) significantly influences its structure. Each lactoferrin molecule contains two N- and C-terminal lobes joined by a helix. Each lobe consists of parts N1 and N2 or C1 and C2. Both lobes possess sites for attaching sugar residues and binding iron. Each of these homologous lobes can attach one iron atom. The forms of LF are classified according to the number of bound iron atoms. LF which attaches iron at each of the two lobes is called holo-LF, whereas the form that has no iron at all is known as apo-LF. There are also intermediate forms which have iron attached at only one lobe. The number of attached iron atoms influences the structure of LF, as the more iron atoms are present, the more cohesive it is, and thus the more resistant to proteolysis and high temperature [3]. Figure 1 shows the form of holo-LF and apo-LF with the marking of the iron-binding site.

Iron binding in hLF leads to conformational changes in the protein structure. The key amino acid to stabilize the Fe(III) ion in the N-lobe of hLF is Arg121. Using the small-angle neutron scattering method, structural differences between the open and closed hLF conformations can be detected. The radius of gyration of apo-LF appears to be smaller than that of holo-LF [10]. The effect of LF iron saturation on its stability under high temperatures was reported by Liu et al. [11]. They showed that the bLF denaturation index in Chinese milk was at least twice as high as in New Zealand milk in the temperature range of 65–80 °C. The authors suggested that this was related to the higher iron content in New Zealand milk (21.9% iron saturation versus 11.6% for Chinese milk). Another study confirmed that LF was more resistant to denaturation when saturated with iron. Binding to iron increases the stability of the protein structure [12]. Regardless of the iron content, LF is stable in the pH range from 4 to 11 [13].

## 4. Acquisition of Lactoferrin

Isolated LF takes the form of a pale pink, odourless powder—Figure 2.

Unfortunately, the isolation of LF is costly, due to the need to process a large volume of whey using chromatographic techniques and ultrafiltration. This is because LF accounts for only 1% of all whey proteins. LF is most often purified using ion-exchange chromatography, which is the greatest expense. This problem was the subject of research by Maciel et al. [1], aimed at reducing the cost of obtaining LF and increasing its recovery from sweet whey. To this end, they used a complex process based on ultrafiltration and expanded bed chromatography. The use of ultrafiltration in the diafiltration mode resulted in a 10-fold decrease in the volume of whey, which was then subjected to chromatography. This increased the degree of its recovery. However, reducing the volume also increases viscosity, which negatively affects the dynamic binding capacity of the bed. Thus, there is a need to establish a concentration limit which cannot be exceeded [1].

Figure 3 presents the LF production process used by the company Morinaga—one of the world leaders in its production. Morinaga LF is produced by a safe method restricting the use of additional chemicals. Only sodium chloride is used, with no buffering salts such as phosphate or acetate. Heat treatment during pasteurization and the freeze-drying of the whey is minimized. In the production process, skim milk or whey is subjected to cation exchange chromatography, followed by elution with sodium chloride. The LF eluate is desalinated and concentrated by ultrafiltration and pasteurized using minimal heat treatment (75 °C, 15 s). After it is freeze-dried and crushed, LF is ready for use in the food or pharmaceutical industries [14,15].

It should be reported that LF is mainly obtained from cow’s milk. However, there is evidence that it differs from hLF in several aspects. Therefore, scientists became interested in the production of recombinant human lactoferrin (rhLF). Currently, it is produced on an industrial scale from *Aspergillus awamori* (Agennix Inc., Houston, TX, USA), rice (Ventria Bioscience, Denver, CO, USA), and transgenic cows (Pharming Group N.V., Leiden, The Netherlands) [16]. On a smaller scale, rhLF can be obtained from plant materials, i.e., yeast, filamentous fungi, tobacco, potatoes, sweet potatoes, tomatoes, ginseng, pears, barley, or wheat [17]. In the animal model, rhLF is produced using, i.a., embryonated chicken eggs, transgenic cows, and goats [18,19,20]. The production of rhLF using transgenic cows is very efficient, as up to 3 g/L can be produced. Moreover, the produced rhLF exhibits the same effect as hLF in terms of the two most important biological activities, i.e., iron binding and iron release [18]. Transgenic goats are able to produce 0.765 mg/mL rhLF with a purity of 97% [21]. Tutykhina et al. [19] attempted to produce rhLF in the allantoic fluid of embryonated chicken eggs. In total, 0.8 mg of protein per embryo was obtained (85% rhLF). The antioxidant and antimicrobial activity of the resulting rhLF was equivalent to that of the hLF [19].

## 5. Effect of Temperature on Lactoferrin Content

Native LF can be supplied to the human body through the consumption of milk and dairy products. However, the origin and heat treatment of milk have been shown to have an enormous impact on the retention of LF in processed dairy products. According to Liu et al. [11], raw milk from New Zealand exhibited 21.9% iron saturation, while milk from China was only 11.6% saturated with iron. This means that the two materials differ significantly in their rate of thermal denaturation of LF. That study assessed thermal processes in the range of 65–121°C in time periods from 2 to 300 s. The selection of these values was deliberate, aimed at representing the thermal processes currently used in milk processing. In the 65–80 °C temperature range, the degree of denaturation of Chinese milk was more than twice the level obtained for New Zealand milk. Interestingly, the denaturation of LF at 85 °C was the same in both materials, but at 65 °C, it was 80 times faster in Chinese milk. At 95 °C, the pattern was reversed, and the denaturation was twice as fast in New Zealand milk as in Chinese milk. The higher stability of LF in New Zealand milk was probably due to its higher iron content. On this basis, the authors developed an Arrhenius model and a kinetic model that can be used to calculate the amount of native LF remaining in the product after heat treatment. In the future, this could be used to design technological processes that will maximize the amount of this valuable protein remaining intact in the product [11].

The denaturation of LF and other proteins also takes place when the protein is exposed to a strong acid, base, or concentrated organic or inorganic salt. The denaturation of the protein changes its structure, which determines its properties such as iron-binding capacity or antibacterial activity. The biological functions of LF can also be influenced by factors such as high pressure, the presence of other proteins and polysaccharides, ionic strength, pH, or temperature. Due to the valuable properties of LF, it is important to retain as much as possible of its native form in the final product. This requires the continual monitoring of the processes and an appropriate selection of processing parameters [3].

## 6. Biological Properties of Lactoferrin

LF has a number of properties positively affecting the human body, mainly anti-inflammatory, antibacterial, antiviral, immunomodulatory, antioxidant, and antitumour properties [1,7,22]. Research by the New Zealand pharmaceutical producer Quantec showed that a protein complex containing LF and lactoperoxidase, obtained from fresh pasteurized cow milk, can protect human cells against COVID-19. The patented defence protein IDP exerts anti-inflammatory, antioxidant, and antibacterial effects [23]. LF has also found its application in tissue engineering [24].

In addition, LF has an anti-allergic effect. The administration of LF reduces allergic skin reactions [25], atopic keratoconjunctivitis [26], and pleuritis caused by hen’s egg albumin [27]. Only one of the studies published so far [28] indicates a potential allergenic effect of LF.

Changes in the lactoferrin gene may also be important in susceptibility to iron deficiency, obesity, or resistance to noroviruses. The human lactoferrin (LTF) gene is located on chromosome 3q [29]. It was confirmed that the expression of the LTF gene in the subcutaneous adipose tissue in people with type 2 diabetes and elevated triglyceride levels is lower than in thin people. This difference is particularly evident in adipocytes [30]. The knockdown of the LTF gene has an effect on the differentiation of human adipocytes, mostly through the modulation of iron homeostasis [31]. In addition, polymorphism in the lactoferrin gene at the position of the 632 T/T genotype is associated with susceptibility to diarrhoea, i.a., in North Americans travelling to Mexico [32].

### 6.1. LF and Obesity

LF exhibits protective activity in obesity. LF supplementation improves body mass index (BMI), waist–hip ratio (WHR), and fasting glucose concentration, as well as insulin sensitivity. LF reduces the accumulation of visceral fat and suppresses appetite, which is particularly important for fighting obesity, a growing problem all over the world. The global incidence of diabetes nearly tripled from 1975 to 2016. In 2016, as many as 39% of adults (over the age of 18) were overweight, and 13% of the global adult population was obese [33].

Table 1 presents the studies showing the potential benefits of introducing LF supplementation in the course or prevention of obesity.

Apart from diet, physical activity and lifestyle, the complex pathogenesis of obesity is significantly influenced by genetic factors [39]. Obesity is a polygenic disease whose pathogenesis involves the FTO, PLIN1, TRAP2B, and BDNF genes [40,41]. The FTO gene encoding the enzyme alpha-ketoglutarate-dependent dioxygenase regulates thermogenesis, energy homeostasis, and the metabolism rate, increases food intake, and takes part in the differentiation of adipocytes, thereby playing an important role in the accumulation of adipose tissue [39,42]. A number of single nucleotide polymorphisms (SNPs) have been identified in this gene, such as rs9939609, rs6499640, rs8050136, and rs1558902, which are positively correlated with the incidence of obesity.

For example, the rs9939609 AA genotype is strongly correlated with the accumulation of adipose tissue. It is associated with the demethylation of ghrelin mRNA N6-methyladenosine, which is linked to the total level of ghrelin and acyl ghrelin. This change results in a weakening of satiety, leading to overconsumption and a preference for high-energy foods [41,42]. Appetite suppression induced by LF intake may prove particularly useful for obese people with the rs9939609 AA genotype. However, this must be confirmed by research in a group selected according to genotype.

Another mechanism of the development of obesity is observed in the case of SNPs located in the perilipin-1 gene (PLIN1). Perilipin is the main protein surrounding the lipid droplet in adipocytes. It modulates interactions between lipids and triacylglycerol reserves [32,33]. Polymorphisms in this gene are associated with the excessive accumulation of adipose tissue and disturbances of cardiometabolic markers [43]. In the case of the rs894160 polymorphism located in the PLIN1 gene, there is a slow but greater reduction in body weight on a diet with a negative energy balance in individuals with the AA variant. In addition, the weight loss lasts longer in people with the AA genotype compared to GG [40]. Holzbach et al. [43] described an interaction of the rs89416 SNP with carbohydrates in the diet, of importance for the dietary treatment of obesity and nutrigenomics. Carbohydrate consumption in people with the PLIN111482 G > A polymorphism modulated waist circumference and the homeostatic model assessment of insulin resistance [43]. The waist circumference increased in carriers of the PLIN 11482G > A GA/AA variant when the intake of complex carbohydrates was <144 g/day. The reverse pattern was observed in carriers of the PLIN 11482 G > A GG variant, in whom a low intake of complex carbohydrates resulted in a more efficient reduction in waist and hip circumference [44]. In terms of nutrigenomics, and thus taking into account the mechanism of action of lactoferrin on the human body as well as biochemical dysfunctions resulting from genotype, lactoferrin can be proposed as a possible nutraceutical of importance not only in the treatment of overweight and obesity but also in their prevention in people with strong predispositions.

A review of the research reveals that LF is an effective component of diet supplements for people with excess body weight, including those with a predisposition for obesity. It is worth noting that overweight or obese individuals often consume dairy products, as they are convenient and do not require preparation. Yoghurt fortified with LF is already being produced, e.g., by the Japanese company Morinaga. In addition, diet supplements containing LF meant for people with excess body weight have already appeared on the market. One example is the Japanese supplement Nice rim essence Lactoferrin^®^ by Lion Corp, recommended in the amount of 3 tablets a day (300 mg lactoferrin) [45].

### 6.2. LF and Type 2 Diabetes and Anti-Inflammatory Effects

Long-term excess body weight and obesity may result in the development of type 2 diabetes. The number of cases of this disease is continually rising. In 2019, diabetes was the ninth most common cause of death. In 2000–2016, there was a 5% increase in premature death due to diabetes [46].

Type 2 diabetes mainly affects adults, but levels of obesity, which is associated with type II diabetes, are rising among children. Mohamed and Schaalan [47] assessed the effect of diet supplementation with LF in a group of 60 children with type 2 diabetes. The children were divided into two groups matched for age and gender. The study participants continued their standard diabetes treatment, but one of the groups additionally received one capsule with 250 mg of camel LF (cLF) every day for three months. The antidiabetic effect of cLF was confirmed by improvements in BMI, lipid profile, and glycated haemoglobin (HbA1c) in the experimental group. There was also a significant decrease in the levels of IL-1β, IL-6, IL-18, TNF-α, and lipocalin-2, which indicates that LF has an anti-inflammatory effect. An antioxidant effect was confirmed by an increase in the level of superoxide dismutase (SOD) and the expression of NrF2. This effect is especially important because obesity is often described as a mild inflammatory disease in which levels of C-reactive protein (CRP), tumour necrosis factor-alpha (TNF-α), and IL-6 increase. The body’s defences are based on an enzymatic antioxidant system as well as less effective endogenous and exogenous antioxidant substances such as vitamin C or uric acid.

Superoxide dismutase (SOD) is an important element of anti-radical defence. The rs4880 polymorphism of the SOD2 gene coding for the important mitochondrial dismutase causes an alanine-to-valine substitution (Ala16Val) [37]. The conformational changes resulting from this mutation lead to a reduction in SOD2 activity. The risk of obesity in people with the Val/Val genotype of rs4880 was shown to be twice as high as in those with the Ala/Ala or Ala/Val genotype [48]. Results obtained by Lewandowski et al. [49] showed that rs4880 variation together with an additional factor such as age causes a 6.19-fold increase in the risk of obesity in individuals with the C/T genotype in comparison with people with the C/C or T/T genotype. In addition, with each year of life, the risk of obesity increased by 5.0%. Interestingly, according to the results of that study, in the model taking into account genotypic variation and BMI, subjects with the C/T genotype had a 5.49-fold lower risk of type 2 diabetes [49]. Thus, it seems that LF intake may be an effective preventive factor and may mitigate the harmful consequences of the rs4880 polymorphism in the SOD2 gene.

The status of antioxidants such as vitamin C in the body is also dependent on genetic predisposition. It should be noted, however, that the role of vitamin C in the body is not limited to its antioxidant properties. The concentration of vitamin C is inversely correlated with insulin resistance and obesity [50]. In addition, vitamin C inhibits lipolysis and the formation and growth of mature adipocytes [51]. Vitamin C is essential for the biosynthesis of carnitine, which is responsible for the transport of long-chain fatty acids through the mitochondrial membrane for the purpose of β-oxidation and the subsequent oxidation of fat. A vitamin C deficiency in the plasma increases the carnitine concentration and thereby limits β-oxidation. People with serum vitamin C deficiencies can thus be resistant to losses of excessive body weight. This situation can be aggravated by dietary behaviour, which can additionally contribute to vitamin C deficiencies. People with excess body weight much less often consume products rich in vitamin C, such as fruits and vegetables [52].

Vitamin C homeostasis in the body can be disturbed by the rs33972313 polymorphism in the SLC23A1 gene, which causes a valine-to-methionine substitution in transporters, leading to a decrease in its concentration in the plasma and halving the rate of accumulation in cells [53]. People with the AA genotype have a 25% lower serum concentration of ascorbic acid, and AG heterozygotes have a 19% lower concentration of ascorbic acid than GG homozygotes [54]. Given the interactions taking place between vitamin C and LF in the human body, their combined supplementation may offer greater health benefits arising from their synergistic activity. This is particularly important in the case of people with polymorphisms limiting the status and bioavailability of vitamin C.

A daily intake of 250 mg of cLF per day confirmed that it exerts anti-inflammatory, immunomodulatory, and hypoglycaemic effects and improves insulin sensitivity. cLF supplementation can thus ensure better control of glycaemia than the use of conventional antidiabetic drugs alone [47]. The possibility of long-term, preventive LF supplementation in people with a genetic predisposition to type 2 diabetes, which is a polygenic disease, should be considered. A number of SNPs associated with diabetes have been identified in various genes. One of the strongly correlated polymorphisms is rs8192678 in the PGC-1α gene. This gene is involved in the homeostasis of energy, lipids, and glucose. As a crucial gene regulating metabolic processes, it is extremely important in the development of obesity, insulin resistance, and type 2 diabetes. Therefore, carriers of the A allele in rs8192678, especially AA homozygotes in the Caucasian and Indian populations, have a significantly higher risk of type 2 diabetes [55]. Long-term hyperglycaemia leads to serious disorders, such as diabetic nephropathy or cardiomyopathy. Hanan et al. [56] conducted a study on an animal model aimed at investigating the effect of quercetin and/or bLF reducing the concentration of advanced glycation end products, connective tissue growth factor, and the expression of inflammatory cytokines in rats with induced diabetic nephropathy and cardiomyopathy. The administration of bLF to rats, especially in combination with quercetin, was effective at controlling kidney and cardiac dysfunction induced by hyperglycaemia [48].

### 6.3. LF and Cardiovascular Diseases

bLF is also used to prevent cardiovascular disease. It exerts a beneficial effect on lipid metabolism and reduces the concentration of total and LDL cholesterol. The positive correlation between a higher serum concentration of bLF and a lower BMI suggests that LF can be used to prevent cardiovascular disease [57].

Dietary intervention based on LF can be one of the corrective measures for patients with the rs268 polymorphism in the lipoprotein lipase gene (LPL). This polymorphism is associated with the content of triglycerides, HDL cholesterol, and total cholesterol in the blood. The GG and AG genotypes are associated with an increased serum triglyceride concentration, which is also positively correlated with age and body weight. In addition, they are associated with an increase in fasting glycaemia and a higher risk of type 2 diabetes and coronary disease [58,59].

It is also worth noting the genetic link between obesity and cardiovascular disease. This association has been noted in studies of polymorphisms of the MTHFR gene. The two most commonly described MTHFR polymorphisms in the literature, C677T (rs1801133) and A1298C (rs1801131), significantly increase the risk of hyperhomocysteinaemia and cardiovascular disease [60]. At the same time, the rs1801133 (C677T) polymorphism is associated with a higher risk of obesity. Studies in animals have confirmed that the administration of LF can reduce the level of homocysteine, thereby lowering the risk of cardiovascular disease [61,62]. Thus, the use of LF is a promising preventive method, especially in people with the TT rs1801133 genotype, in whom the homocysteine concentration is usually 2.91 mmol/L higher than in people with the CC variant. It should be noted that the odds ratio (OR) for obesity was 1.23 for a 5 μmol/L increase in the homocysteine concentration [62].

The concentration of circulating LF in the blood is also inversely proportional to the level of free fatty acids following the use of a high-fat diet. In addition, LF exhibits strong antihypertensive activity, as it blocks angiotensin receptor AT1 and inhibits the renin-angiotensin-aldosterone system [63]. Ling et al. [57] conducted a study to test the effect of bLF supplementation on the development of atherosclerosis in mice. The mice were divided into three groups, one of which received distilled water (HFCD), whereas the other two received a solution of bLF at a concentration of 2 mg/mL (MLF) or 10 mg/mL (HLF). In comparison with group HFCD, the mice in groups MLF and HLF had much lower total cholesterol levels in the blood serum (MLF by 40%, HFL by 27.7%). In addition, there was a significant increase in the excretion of cholesterol in the faeces, by 53.4% in group MLF and by 40.8% in group HLF. bLF mitigated fatty liver and lesions in the aorta in the MLF and HLF mice. There was also a reduction in the expression of the protein HMG-CoA reductase, which reduces cholesterol synthesis, and an increase in the activity of cholesterol 7-alpha hydroxylase, an enzyme which slows down the synthesis of bile acids from cholesterol.

LF is able to bind iron, and for this reason, it has been used to treat anaemia or iron deficiency. One of the predisposing factors for a low iron concentration in the body is genetic variation. A number of polymorphisms associated with serum levels of iron have been identified. One of these is the single nucleotide polymorphism rs855791, located in the gene encoding the enzyme transmembrane serine protease 6 (TMPRSS6). This enzyme is associated with the absorption and resorption of iron in the body. The presence of the T allele reduces the iron concentration in the serum, the saturation of transferrin with iron, haemoglobin content in the erythrocytes, and mean corpuscular volume [64]. Bovine LF has been shown to be very similar to human LF in its structure and function. Therefore, it can be effective at regulating inflammation (it reduces levels of IL-6) and increasing iron levels in the body, as well as restoring its physiological transport to cells [65]. In addition, LF supplementation, compared to ferrous sulphate, does not cause side effects. The absence of side effects from bLF is most likely linked to the lack of free iron available in the gastrointestinal tract. The presence of free iron disturbs intestinal peristalsis and irritates the mucosa, causing adverse gastrointestinal effects. During supplementation with free iron salts, only 20–30% of the dose is absorbed, and the remainder causes gastrointestinal dysfunction [66].

Table 2 presents the studies showing the effect of LF supplementation on haematological parameters.

### 6.4. LF and Immune System

Human LF, hLF, also exerts a positive effect on the immune system, as it modulates both specific and nonspecific immune mechanisms. hLF promotes the maturation, differentiation, and activation of T and B lymphocytes. The introduction of lactoferrin to the body in the diet can influence the systemic immune response by modulating the activity of cytokines (IL-4, IL-2, and IL-12) and their expression in the blood and lymph [70]. Because it is present in body fluids, i.e., saliva, tears, mucous, and gastrointestinal fluids, it is the first line of defence against infections [71,72,73]. This includes bacterial infections (*Escherichia coli* [74,75], *Helicobacter pylori* [76], and *Staphylococcus aureus*) [75], viral infections (herpes virus (HSV) [77], human immunodeficiency virus (HIV) [78], hepatitis type B and C [79] virus (HBV and HCV), or rotaviruses), and fungal infections (*Candida albicans*) [73]. LF reduces inflammation and has beneficial effects in cancer patients, exhibiting cytotoxic activity through damage to cell membranes and cell lysis, which reduces tumours [80]. hLF is utilized by immune cells and supplied locally in the case of inflammation by neutrophils, which store it in their secondary granules. Following the activation of neutrophils, hLF is released into the blood, where its concentration can increase from about 1 mg/mL (level in normal conditions) to even 200 mg/L. Due to its high affinity for iron, hLF is able to capture iron from the surface of microorganisms, thereby limiting their growth and protecting the target host cells. In addition, hLF forms complexes with proteins taking part in the immune response, such as osteopontin. These are thought to be able to modulate the antibacterial and immunostimulatory effects of hLF [70].

Anti-inflammatory, anti-infective, and immunoregulatory agents have been shown to be useful in treating infections with SARS-CoV-2. Lactoferrin exhibits all of these effects and can be included in the treatment of SARS-CoV-2 infection. LF has been shown to exhibit activity against a wide range of viruses, including SARS-CoV, which is closely related to SARS-CoV-2 [81,82]. LF can inhibit the binding of SARS-CoV-2 to host cells. LF may therefore be beneficial in preventing the occurrence of SARS-CoV-2 [83]. The administration of LF preparations to patients with COVID-19 may be effective at controlling gastrointestinal symptoms and may limit the spread of the virus from the faeces and intestines. LF can be used as a primary or supplementary treatment in both symptomatic and asymptomatic patients [84]. Quantec noted that the use of a complex of milk proteins yielded better results against COVID-19 than pure LF or lactoperoxidase. The company patented the protective milk-derived protein IDP. This complex strengthens the immune system and therefore is effective against influenza and herpes and may be helpful in the fight against SARS-CoV-2 [85].

Another valuable function of LF is its capacity to reduce oxidative stress. During inflammation, phagocytes release reactive oxygen species, which neutralize microorganisms. Unfortunately, reactive oxygen species also negatively affect live tissue, leading to necrosis. Tissues then release iron, which takes part in the Haber–Weiss reaction, leading to the generation of more free radicals. LF, owing to its iron-binding capacity, helps to alleviate oxidative stress [70,86].

The composition of the gut microbiota is an important element of the immune system. Bifidobacteria, which are naturally present in the human gastrointestinal tract, exert a positive influence on the immune response and prevent infection. Bifidobacteria are more often present in the gastrointestinal tract of breastfed infants than in those receiving formula because human milk, and especially colostrum, is naturally rich in LF. The LF concentration in human colostrum is 5.9 mg/L. Over the course of lactation, it falls to 2.9 mg/L in transitional milk and to 2.5 mg/mL in mature milk [57,58].

Table 3 presents the studies on the effect of LF supplementation on the immune system.

## 7. Scale of Lactoferrin Production

Due to its unique properties, interest in LF is continually growing. The demand for the protein nearly doubled in the years 2014–2020. A key producer in Europe—Friesland Campina Ingredients—is planning to increase production to 70 tonnes this year in response to the growing market demand [91]. Similar steps are being taken by other world producers, including Morinaga, whose German factory declared an increase in production capacity to 170 tonnes [92]. The global bLF market in 2021 was valued at 235.9 million USD, and this value is projected to increase to 411.8 million USD in 2028 [93]. The European lactoferrin market was worth 15.37 million USD in 2021 and is expected to increase to 23.53 million USD by 2026 [94]. Key LF producers have their factories mainly in Europe, New Zealand, Australia, and California. The exportation of LF to other parts of the world, e.g., South America, is one of the factors affecting the high price of LF [95].

Table 4 presents a list of key lactoferrin producers around the world.

## 8. Applications in Industry

LF is used in the food sector, pharmaceutical industry, and cosmetics industry. It is used in infant formulas, nutraceuticals, fermented milk products, processed meats, and dietary supplements [16,97]. In the pharmaceutical and cosmetics industries, it appears as an ingredient in preparations supporting the treatment of herpes or skin lesions, strengthening immunity, raising iron levels, supporting the intestinal microbiota, or limiting bacterial growth [73].

Table 5 presents selected pharmaceutical and cosmetic products manufactured using LF.

In the European Union, bLF has been approved for use as a food ingredient since 2012, following the registration of the protein as a novel food by Friesland Campina. In the application for infants aged 0–6 months, the bLF intake is set at 200 mg per kg bodyweight and 1.2 g per day. The recommendation for adults is at the level of 1.4–3.4 g per day. Friesland Campina presented a mouse study that showed no adverse effects at 2000 mg/kg of body weight. LF can be used in various categories of food. The amounts that have been approved for use are presented in Table 6. Foods to which LF has been added must have ‘lactoferrin from cows’ milk’ written on the label [14,98].

Products containing LF are already available in the food sector. The widest assortment is available for powdered milk for infants, but other products have appeared as well, such as chewing gum, jelly sweets, non-alcoholic beverages, and yoghurt (Table 7).

The use of LF in infant formulas has many beneficial health-promoting effects, increasing immunity and positively influencing the skeletal and digestive systems [97]. Such formulas are mainly found in Spain, Indonesia, and South Korea [16]. Li et al. [100] assessed the effect of modified milk enriched with bLF in the amount of 0.6 g/L and bovine milk fat globule membrane (MFGM) on infant development. The results of the study show that children receiving this mixture had a higher neurodevelopmental profile at 365 days of age and higher language abilities at 545 days. Moreover, diarrhoea, vomiting, fungal infections, and respiratory problems were less common in these children.

The use of LF is also of technological importance. LF digested with pepsin–lactoferricin is a strong antimicrobial peptide that can be used in food preservation due to its high resistance to high temperatures [101]. LF in yoghurt production positively affects the physicochemical structure of the product. In this case, however, the form of lactoferrin is significant, as apo-LF has been shown to inhibit the growth of lactic acid bacteria, while holo-LF had the opposite effect. The use of LF had no negative effect on the shelf life of yoghurt [97]. It is especially beneficial to add this protein to yoghurt, and to a lesser extent to other dairy products as well. The use of bLF in yoghurt affects not only its physicochemical properties but also the health of its consumers. Tsukahara et al. [102] conducted a study in a group of 578 nursery school children, who were given yoghurt containing 100 mg LF for 15 weeks. The control group comprised 584 children who received fruit jelly instead of yoghurt. The children receiving yoghurt with LF ≥ 3 days a week were much less often absent from school due to vomiting (4.3%) than children from the control group (8.4%).

LF is used in the meat industry for its antibacterial properties. An LF suspension not exceeding 0.20 mL/kg can be sprayed onto the surface of meat. This technique can be used just before packaging the meat to increase its shelf life. This discovery has raised interest in LF as a component of food packaging [97]. Barbiroli et al. [103] assessed the effect of LF on meat storage by adding it to cellulose packaging, which was tested on thin cuts of meat. The paper was produced with LF in the amount of 10% of the total fibre content. The addition of LF to paper packaging inhibited the growth of harmful bacteria, which in the future could be exploited in absorbent pads or films for wrapping meat [80]. Another study evaluating the impact of LF on the length of storage of raw meat was conducted by Soyer et al. [104]. The results confirmed the antibacterial effect of active lactoferrin on *L. monocytogenes* and *E. coli*. Padrão et al. [105] used cellulose film to which LF was added. The film was considered edible packaging and exhibited antimicrobial properties. It could be used to store perishable food, such as fresh sausage. The films with LF inhibited *E. coli* and *S. aureus* bacteria. However, it can be difficult to add lactoferrin to edible packaging, because too much of it can negatively affect the structure of the material. The addition of LF to pullulan films at low concentrations, i.e., <0.03%, did not significantly affect their tensile strength or elongation at break, but higher concentrations negatively affected these parameters. It is important to preserve the durability of the material while also using a concentration sufficient to exploit its ability to inhibit microbial growth [106].

## 9. Conclusions

LF is a protein with multiple health-promoting properties. It exhibits anti-inflammatory, antibacterial, antiviral, immunomodulatory, antioxidant, and anti-tumour activity. The LF consumption from dairy products is insufficient to observe additional health benefits. Therefore, the supplementation or fortification of food products with LF is recommended. Supplementation with LF has a protective effect in obesity, positively influencing BMI and WHR and suppressing appetite. Moreover, LF can be used as a supportive treatment in diabetes by improving glycaemia and reducing inflammation. In addition, in cardiovascular diseases, LF has an antihypertensive effect and improves lipid parameters. Due to its ability to bind iron, it has also been used to treat iron deficiency and anaemia. All of these properties have led to the increasingly common use of LF as an ingredient in pharmaceuticals and cosmetics. It is also added to food products, such as modified milk, yoghurt, chewing gum, and even jelly sweets. Thanks to its properties, lactoferrin is used not only as a bioactive component of the diet. There are also many studies showing its use as a functional ingredient, e.g., as a packaging ingredient, which exhibits natural antiseptic properties.

The properties of LF also allow it to be widely used in diet therapy based on evidence-based medicine (EBM) and nutrition genomics. The use of nutrigenomics and nutrigenetics tools allows to identify a number of single nucleotide polymorphisms in such genes as, for example, FTO, PLIN1, TRAP2B, BDNF, SOD2, SLC23A1, LPL, and MTHFR. The knowledge of a person’s genetic suit allows for personalized nutritional prophylaxis, the task of which is to reduce the identified risk of diseases, particularly in the case of already existing diseases associated with a specific genetic predisposition to conduct targeted diet therapy. The use of nutrition genomics allows for the effective stratification of people and the selection of the most optimal bioactive nutrients, including LF, whose bioactive potential cannot be considered without taking into account the group to which they will be given.

## Figures and Tables

**Figure 1 foods-12-00070-f001:**
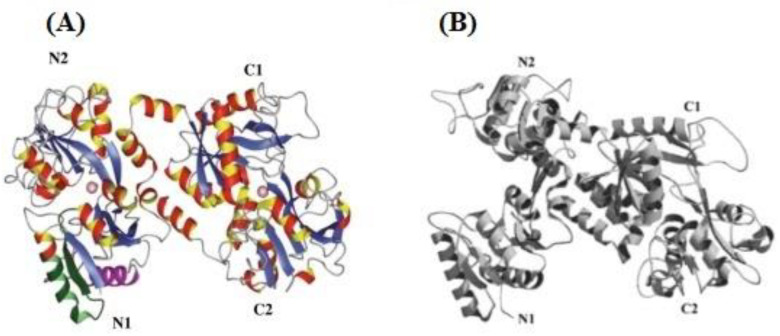
(**A**) Structure of holo-LF (pink spheres represent the iron binding site). (**B**) Structure of apo-LF [3].

**Figure 2 foods-12-00070-f002:**
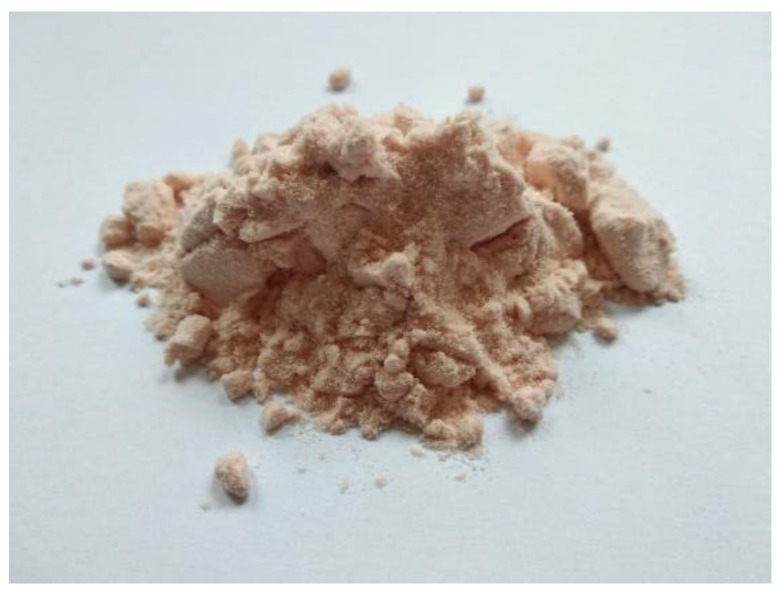
LF isolated from milk [phot. A. Jańczuk].

**Figure 3 foods-12-00070-f003:**
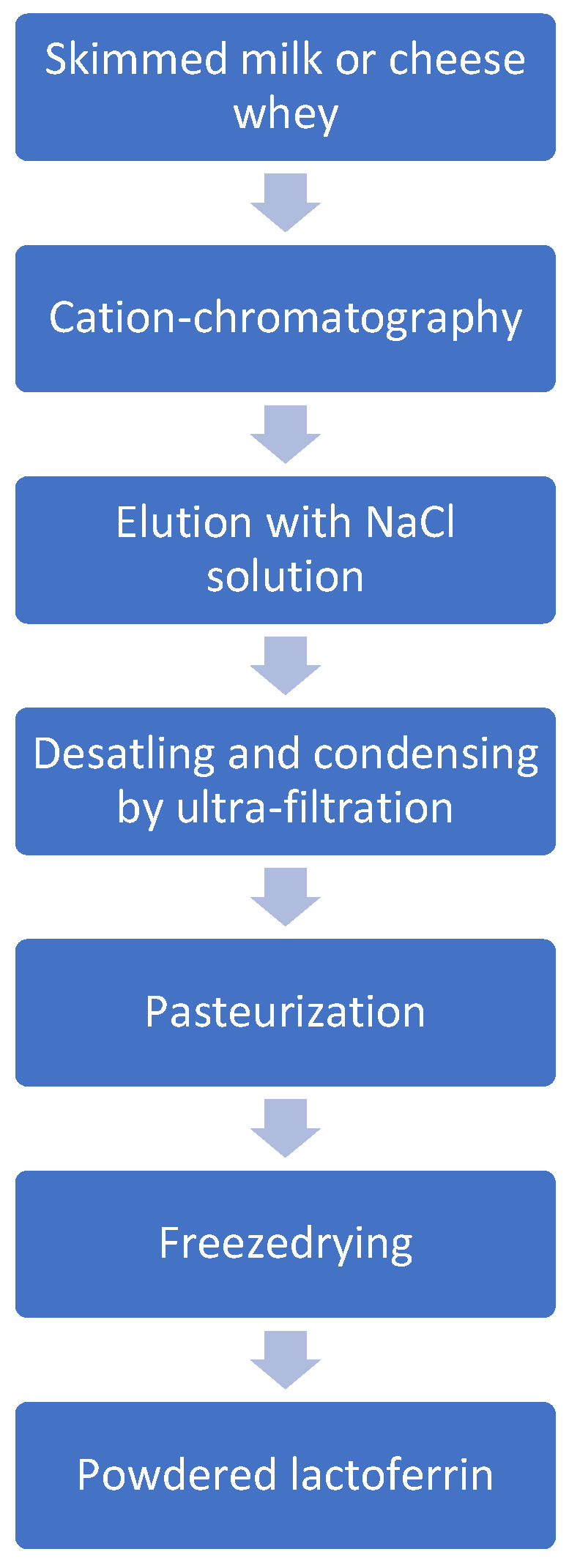
LF production process.

**Table 1 foods-12-00070-t001:** Use of LF supplementation in obesity (own work).

LF Form	Dosage	Research Assumptions	Effect of LF	References
bLF	300 mg	Effects of bLF on systemic lipid metabolism and visceral fat reduction	A significant reduction in visceral fat area (VFA) was observed (−14.6 cm^2^), as well as a decrease in body weight (−1.5 kg), BMI (−0.6 kg/m^2^) and waist circumference (−4.4 cm)	[34]
Enteric-coated LF (eLF)	No data	Effect of LF supplementation on weight reduction (excluding subjects with BMI ≥ 30 and/or hyper-LDL cholesterolemia)	eLF caused a significant decrease in adipose tissue (−10.2 cm^2^, *p* < 0.01) and reduction in the average visceral fat content (−16.2 cm^2^, *p* < 0.01) and a significant decrease in BMI (−0.38 kg/m^2^, *p* < 0.05)	[35]
LF	200 mg/day	Effect of LF on weight loss and appetite in obese school-age children	LF was shown to reduce appetite in obese school-aged children. It did not cause weight loss	[36]
bLF	100 mg/kg BW	Effect of LF supplementation on the development of obesity on a high-fat diet (HF) in mice	bLF reduced the increase in body weight gain by 50%, decreased the total cholesterol concentration in the blood serum, and positively influenced the blood glucose level. No differences were noted in triacylglycerol content	[37]
LF	15% of total protein source	LF prevents obesity and diet-induced development formation of hepatic steatosis in C57Bl/6J mice	LF supplementation was shown to improve weight loss and limit its subsequent regain, mitigated fatty liver, improved glucose tolerance, and reduced the inflammation of adipose tissue	[38]

**Table 2 foods-12-00070-t002:** The effect of LF supplementation on iron balance in the body (own work).

LF Form	Dosage	Research Assumptions	Effect of LF	References
bLF	200 mg/day	Effectiveness of bLF and ferrous sulphate in treating iron deficiency and iron-deficiency anaemia in pregnant women	Administration of 30%-iron-saturated bLF to pregnant women increased haemoglobin levels, erythrocyte counts, and levels of ferritin and total iron. bLF caused a decrease in the IL-6 level and an increase in the hepcidin concentration	[67]
bLF	200 mg/day	Safety and efficacy of LF versus ferrous sulphate in pregnant women	Supplementation LF decreased the IL-6 level by 26% but increased by 21% in the group that received ferrous sulphate. No side effects were noted in the LF group, whereas in the ferrous sulphate group, they were reported by 16.5% of women	[65]
bLF	38 mg bLF/100 g infant formula	Efficacy of lactoferrin-enriched infant formula in improving hematologic indices and iron status	The body weight of the infants receiving bLF was significantly higher than that in the group that received iron. Milk enriched with bLF significantly increased the concentration of haemoglobin (Hb) by 12%, ferritin by 76%, and total iron by 29%, significantly decreasing the detection rate of anaemia from 12% to 5% and of iron-deficiency anaemia from 35% to 16%	[68]
bLF	200 mg/day	Evaluation of the effectiveness of LF administration before and after a meal. The possible influence of digestive protease on LF lactoferrin degradation	An increase in serum haematological parameters, i.e., haemoglobin, erythrocytes, ferritin and iron, was observed only in the group in which the bLF supplement was administered before meals	[69]

**Table 3 foods-12-00070-t003:** Effect of LF supplementation on the immune system (own work).

LF Form	Dosage	Research Assumptions	Effect of LF	References
bLF	Placebo—first week 100 mg—second week 200 mg—third week	Potential immune-modulating properties and antioxidant activity of oral supplementation of bovine lactoferrin in humans.	After two weeks of bLF supplementation, the total T cell count had significantly increased by 29%, helper T cells by 22%, and cytotoxic T cells by 25%. The number of NK cells was not affected by the supplementation. An important finding was the significant increase in antioxidant capacity after two weeks of LF supplementation	[87]
Liposomal bovine LF (LLF)	128–192 mg/day	LLF as potential prevention and cure for COVID-19.	Administration of LLF led to 100% recovery in 75 SARS-CoV-2-positive patients within 4–5 days	[88]
LLF	1 g/day	Evaluation of the antiviral effect of oral and nasal LLF in asymptomatic and mild to moderate COVID-19 patients.	Faster recovery from COVID-19 in patients taking LLF	[89]
bLF	0.6 g/L	Comparison of the gut microbiome profile and its metabolites in infants up to 12 months of age who received formula containing bLF and MMGF (milk fat globules) or regular formula without these additives.	Infants fed the formula with bLF and MMGF showed an increase in numbers of Bacteroides uniformis and Bacteroides plebeius and of bifidobacteria in the faeces, accompanied by a decrease in numbers of E. coli, in comparison to infants receiving the standard formula up to 4 months of age	[90]

**Table 4 foods-12-00070-t004:** Key lactoferrin producers around the world (own work based on [96]).

Producer	Country
MILEI GmbH	Leutkirch im Allgäu, Germany
FrieslandCampina DOMO	Amersfoort, The Netherlands
Synlait Ltd.	Dunsandel, New Zealand
Glanbia Plc.	Kilkenny, Ireland
Bega Bionutrients	Port Melbourne, VIC, Australia
Saputo Dairy Australia Pty Ltd.	Allansford, VIC, Australia
Fonterra Co-operative Group	Auckland, New Zealand
Armor Protéines SAS	Loudéac, France
Hilmar Cheese Co.	Hilmar, CA, USA
Murray Goulburn Co-operative Co. Ltd.	Southbank, VIC, Australia
Ingredia	Arras, France

**Table 5 foods-12-00070-t005:** Use of LF in pharmaceutical products and cosmetics (own work).

Product Type	Product Name	Lactoferrin Content	Use	Web Page Link
Cream	Endvir Simplex (Vitis Pharma)	6%	- treatment of herpes - treatment of mucosal lesions	http://vitispharma.pl/katalog/produkt/produkty/endvir/endvir-simplex (accessed on 16 May 2022)
Cream	Acnex (Farmina Ltd.)	No data	- treatment of acne	https://farmina.pl/product/acnex-krem/ (accessed on 16 May 2022)
Capsules	Lactoferrin (Pharmabest)	100 mg/capsule	- enhancing immunity - supporting treatment of colds and infections	https://pharmabest.pl/sklep/produkt/laktoferyna-kapsulki/ (accessed on 16 May 2022)
Capsules	Lactoferrin (Jarrow Formulas)	250 mg/capsule	- supporting intestinal microbiota - mitigating symptoms of lactose intolerance - treatment of *Helicobacter pylori* - stimulation of the immune system - improvement of iron absorption	https://jarrow.com/products/lactoferrin-250-mg-60-capsules (accessed on 16 May 2022)
Capsules	IronSorb + Lactoferrin (Jarrow Formulas)	200 mg/capsule	- correction of iron deficiency	https://jarrow.com/products/ironsorb-lactoferrin-60-capsules (accessed on 16 May 2022)
Capsules	Lactoferrin LFS 90% 100 mg (Aliness—MedicaLine)	100 mg/capsule	- supporting immunity	https://aliness.pl/pl/p/Lactoferrin-LFS-90-100-mg-x-30-kapsulek/228 (accessed on 16 May 2022)
Oral drops	Lactoferrin (Pharmabest)	100 mg/12 drops	- supporting immunity - supporting treatment of infections - correcting iron imbalances - regeneration of intestinal epithelium (bifidogenic activity)	https://pharmabest.pl/laktoferyna-krople-doustne/ (accessed on 16 May 2022)
Sachets	Lactoferrin (Pharmabest)	100 mg/sachet	- support in immunodeficiency - support in iron deficiency - reducing the risk of sepsis in premature infants with very low birth weight - stimulation of the growth of beneficial gut microbiota - antioxidant activity	https://pharmabest.pl/sklep/produkt/laktoferyna-saszetki/ (accessed on 16 May 2022)
Sachets	Fibraxine (Alfasigma)	50 mg/sachet	- supporting physiological intestinal function	https://www.alfasigma.com/ (accessed on 16 May 2022)
Dragées	Fiorda Junior (PhytoPharm)	6 mg/dragée	- treatment of irritation of the throat and larynx	http://phytopharm.pl/en/fiorda (accessed on 16 May 2022)
Toothpaste	Lactoferrin Toothpaste (DENTE91)	No data	- tooth regeneration - inhibition of growth of bacteria inducing gingivitis - prevention and treatment of caries	https://dente91.com/dente91 (accessed on 16 May 2022)
Mouthwash	Lactoferrin Mouthwash (DENTE91)	No data	- maintaining oral hygiene - protection against gingivitis, periodontitis, oral mycosis, and bacterial infections - prevention of tartar and plaque formation	https://dente91.com/dente-91 (accessed on 16 May 2022)

**Table 6 foods-12-00070-t006:** Use of bLF in food products [99].

Food Category	Maximum Level of bLF
Infant formulae and follow-on formulae within the meaning of Regulation (EU) No 609/2013 (ready to drink)	100 mg/100 mL
Dairy-based foods for small children (ready-to-eat)	200 mg/100 g
Processed cereal products (solid)	670 mg/100 g
Food for special medical purposes within the meaning of Regulation (EU) No 609/2013	Up to 3 g/day
Milk-based beverages	200 mg/100 g
Powdered milk-based drink mixes (ready-to-drink)	330 mg/100 g
Beverages based on fermented milk (including yoghurt drinks)	50 mg/100 g
Non-alcoholic drinks	120 mg/100 g
Yoghurt-based products	80 mg/100 g
Cheese-based products	2000 mg/100 g
Ice cream	130 mg/100 g
Cakes and pastries	1000 mg/100 g
Candies	750 mg/100 g
Chewing gum	3000 mg/100 g

**Table 7 foods-12-00070-t007:** Use of LF in food products (own work).

Product Type	Product Name	Lactoferrin Content	Country of Manufacture	Use	Web Page Link
Modified powdered milk (up to 1 year of age)	Morinaga Hagukumi (Morinaga Milk Industry Co., Ltd)	80 mg/100 g powder	Tokio, Japan	- feeding new-borns and infants - improves resistance - supports gut microbiota	https://www.morinagamilk.co.jp/english/products/jp/infantformula.php (accessed on 17 May 2022)
Modified powdered milk (up to 1 year of age)	Enfamil Enspire Infant Formula (Mead Johnson & Company)	<2%	Chicago, IL, USA	- feeding new-borns and infants - reduces the risk of gastrointestinal and respiratory disease	https://www.enfamil.com/products/enfamil-enspire-infant-formula/ (accessed on 17 May 2022)
Mleko Modified powdered milk (from 12 months to 3 years of age)	Morinaga Chil-mil (Morinaga Milk Industry Co., Ltd)	55 mg/100 g powder	Tokio, Japan	- supplementary nutrition for small children - raises iron level in the body - supports immunity	https://www.morinagamilk.co.jp/english/products/jp/infantformula.php (accessed on 17 May 2022)
Yoghurt	Lactoferrin Yogurt (Morinaga Milk Industry Co., Ltd)	100 mg	Tokio, Japan	- no data	https://www.morinagamilk.co.jp/english/products/jp/yogurt.php (accessed on 17 May 2022)
Chewing gum with lactoferrin and manuka honey	Blue^®^m dental chewing gum (Blue^®^m)	No data	VH Wijhe, Netherlands	- protection against fungi, bacteria, and viruses - supplements daily oral hygiene	https://bluemcare.com/product/dental-chewing-gum/ (accessed on 17 May 2022)
Carbonated drink with lactoferrin	Immune + (Beston Global Foods)	No data	Adelaide, SA, Australia	- supports immunity	https://immuneplus.com.au/ (accessed on 17 May 2022)
Fruit gums for children	C + Zinc Jelly (NANA and TAKA)	No data	Taiwan	- supports immunity	https://www.hktvmall.com/hktv/en/main/Hong-Kong-Chien-Cao-Tong-Medical-Limited/s/H6449002/Personal-Care-%26-Health/Personal-Care-%26-Health/Health/Immunity/CZinc-Jelly/p/H6449002_S_CCT10867?scrollTo=recommendationTab (accessed on 17 May 2022)

## Data Availability

Not applicable.

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
