# Peer review of "Lactoferrin—The Health-Promoting Properties and Contemporary Application with Genetic Aspects"

_foods, 2022, doi:10.3390/foods12010070_

Round 1

Reviewer 1 Report

- The article presents a detailed review of the benefits of lactoferrin not only on classic diseases such as diabetes and dilipidemia, but also providing updates on the impact on COVID-19.

- Table 1 shows the world's largest lactoferrin producers, basically covering the European continent and Australia, with Latin America represented by the United States of America. Does this mean that all lactoferrin sold in other countries, such as South America, is imported? what would justify its high cost, in addition to the already demonstrated extraction costs?

- Reports in the text show an average consumption of 250mg of lactoferrin with effects in the reduction of obesity and beneficial effects in type 2 diabetes. the amount. Thus, this observation should be highlighted in the conclusion, that despite all these benefits, there is still a deficit if lactoferrin is ingested only in dairy products, without supplementation.

Author Response

Dear Reviewer 1,

Thank you for giving us the opportunity to submit a revised version of our manuscript titled Lactoferrin – the Health-Promoting Properties and Contemporary Application with Genetic Aspects” to Foods.

We appreciate the time and effort that you have dedicated to providing your valuable feedback on our manuscript. We are grateful for your comments on our paper. We have made corrections in the text in accordance with the suggestions.

Reviewer: 1)     Table 1 shows the world's largest lactoferrin producers, basically covering the European continent and Australia, with Latin America represented by the United States of America. Does this mean that all lactoferrin sold in other countries, such as South America, is imported? what would justify its high cost, in addition to the already demonstrated extraction costs?

Authors: Thank you for this suggestion. It has been corrected in the text (L534-536).

Import is another factor contributing to the high cost of lactoferrin. The companies with the highest LF sale profits in South America come from Canada, England and New Zealand.

Reviewer: 2)     Reports in the text show an average consumption of 250mg of lactoferrin with effects in the reduction of obesity and beneficial effects in type 2 diabetes. the amount. Thus, this observation should be highlighted in the conclusion, that despite all these benefits, there is still a deficit if lactoferrin is ingested only in dairy products, without supplementation.

Authors: Thank you for this suggestion. It has been corrected in the text (L615-617).

Modified lines: 615-617 “The LF consumption from dairy products is insufficient to observe additional health benefits. Therefore, the supplementation or fortification of food products with LF is recommended.”

Sincerely,

Authors

Reviewer 2 Report

-The review article focused on the importance of LF in food industry and reviewed how LF plays an important role in diabetes, cardiovascular diseases, and other systemic diseases. This review article will help the readers to understand further the importance of LF for alternate to many pharmaceutical approaches. However, the manuscript requires Extensive English editing to improve the quality of manuscript.

-Authors should mention whether human lactoferrin or bovine lactoferrin or recombinant lactoferrin. Also, authors should add many clinical studies were carried out of various sources of LF.

-The tables should be quote appropriate references for eg. Table 2.

-The authors mentioned in the table as own work. The authors should add references if it was published or add web page link so that the readers can directly click the link and read if they want to get more information on that study.

-Authors should add references to quote other studies in the manuscript

for e.g., the following should add references;

"Because it is present in body fluids, i.e., saliva, tears, mucous, and gastrointestinal fluids, it is the first line of defense against infections. This includes bacterial infections (Escherichia coli, Helicobacter pylori and Staphylococcus aureus), viral infections (herpes virus (HSV), human immunodeficiency virus (HIV), hepatitis type B and C virus (HBV and HCV), or rotaviruses), and fungal infections (Candida albicans)".

-Most of the paragraphs are lengthy need to reduce.

-Authors should add the limitation of the LF usage, side effects and if any allergic reaction etc.,

-The authors concluded that the importance of LF for personalized medicine. There are several SNPs are identified in the LF gene and was documented the importance of each SNPs. So, the authors should add LF polymorphism and its activity on antibacterial, anti-viral and anti-inflammatory activities and various diseases. 

Author Response

Dear Reviewer 2,

Thank you for giving us the opportunity to submit a revised version of our manuscript titled Lactoferrin – the Health-Promoting Properties and Contemporary Application with Genetic Aspects” to Foods.

We appreciate the time and effort that you have dedicated to providing your valuable feedback on our manuscript. We are grateful for your comments on our paper. We have made corrections in the text in accordance with all the suggestions.

Reviewer: Authors should mention whether human lactoferrin or bovine lactoferrin or recombinant lactoferrin.

Authors: Thank you for this suggestion. It has been corrected in the text (Lines: 192, 195, 212, 215, 220, 222, 283, 284, 328, 329, 341, 344, 348, 350, 375, 377, 381, 397, 407, 412, 414, 428, 429,-434, 442, 445, 450, 451, 454-481, 496, 531).

Reviewer: Also, authors should add many clinical studies were carried out of various sources of LF.

Authors: Thank you for this suggestion. It has been included and completed in the text of manuscript (L164-534).

Reviewer: The tables should be quote appropriate references for eg. Table 2.

Authors: Thank you for this suggestion. Reference to Table 3 has been corrected (L559).

Reviewer: The authors mentioned in the table as own work. The authors should add references if it was published or add web page link so that the readers can directly click the link and read if they want to get more information on that study.

Authors: Thank you for this suggestion. Web page links have been added to Tables 2 and 4.

Reviewer: Authors should add references to quote other studies in the manuscript

for e.g., the following should add references;

"Because it is present in body fluids, i.e., saliva, tears, mucous, and gastrointestinal fluids, it is the first line of defense against infections. This includes bacterial infections (Escherichia coli, Helicobacter pylori and Staphylococcus aureus), viral infections (herpes virus (HSV), human immunodeficiency virus (HIV), hepatitis type B and C virus (HBV and HCV), or rotaviruses), and fungal infections (Candida albicans)".

Authors: Thank you for this suggestion. References have been added (L458-465).

Reviewer: Most of the paragraphs are lengthy need to reduce.

Authors: Thank you for this suggestion. Paragraphs in the text have been shortened (L109, 177, 190, 223, 230, 263, 278, 292, 304, 337, 344).

Reviewer: Authors should add the limitation of the LF usage, side effects and if any allergic reaction etc.,

Authors: Thank you for this suggestion. It has been included and completed in the text of manuscript.

The article presents limits for the use of bovine lactoferrin in food products (L526-536). Recommended intake doses and amounts at which no adverse effects were observed have been added in Lines 554-558. At the moment, there are no references directly showing side effects related to the consumption of lactoferrin. LF is considered as safety by EFSA. In many parts of the text the limitations of usage are presented, e.g. L 121-124, 191-203, 204-211, 238-244, 245-249, 278-292, 293-298, 307-310, 319-327, 348-351, 352-357, 359-370. Only one of the studies published so far [Negaoui et al. 2016] indicates a potential allergenic effect of LF (L170-173).

Reviewer: The authors concluded that the importance of LF for personalized medicine. There are several SNPs are identified in the LF gene and was documented the importance of each SNPs. So, the authors should add LF polymorphism and its activity on antibacterial, anti-viral and anti-inflammatory activities and various diseases. 

Authors: Thank you for this suggestion. Information about the LF gene and its links with disease entities has been completed (L 174-182).

We hope the answers are comprehensive.

Sincerely,

Authors

Reviewer 3 Report

This is an excellent and comprehensive survey of the chemical and health properties of lactoferrin, with novel insights pertaining to the need to consider the genomic profile of potential consumers of lactoferrin and other bioactive nutrients.

I would make only one suggestion: some further detail should be provided on the structure of lactoferrin, particularly on the possible roles of iron in the stability of the protein under variable heat, pH or ionic concentration conditions. Although the authors already discuss this in the text (section 3. Occurrence and structure of lactoferrin), additional figures illustrating the overall structure of lactoferrin, specific interactions between iron chelating sidechains and ferric/ferrous iron, and conformational differences in apo and holo forms of LF would facilitate a better understanding of the biochemical influences underlying structural properties and stability.

Author Response

Dear Reviewer 3,

Thank you for giving us the opportunity to submit a revised version of our manuscript titled Lactoferrin – the Health-Promoting Properties and Contemporary Application with Genetic Aspects” to Foods.

We appreciate the time and effort that you have dedicated to providing your valuable feedback on our manuscript. We are grateful for your comments on our paper. We have made corrections in the text in accordance with all suggestion, i.e.:

Reviewer: some further detail should be provided on the structure of lactoferrin, particularly on the possible roles of iron in the stability of the protein under variable heat, pH or ionic concentration conditions.

Authors: Thank you for this suggestion. Authors has referred to these suggestions in the text (L90-101).

Reviewer: Although the authors already discuss this in the text (section 3. Occurrence and structure of lactoferrin), additional figures illustrating the overall structure of lactoferrin, specific interactions between iron chelating sidechains and ferric/ferrous iron, and conformational differences in apo and holo forms of LF would facilitate a better understanding of the biochemical influences underlying structural properties and stability.

Authors: Thank you for this suggestion. Figure 1 has been added to the text, showing the form of holo-LF and apo-LF with the marking of the iron-binding site (L85-89).

Sincerely,

Authors